# Entropy Measurements for Leukocytes’ Surrounding Informativeness Evaluation for Acute Lymphoblastic Leukemia Classification

**DOI:** 10.3390/e24111560

**Published:** 2022-10-29

**Authors:** Krzysztof Pałczyński, Damian Ledziński, Tomasz Andrysiak

**Affiliations:** Faculty of Telecommunications, Computer Science and Electrical Engineering, Bydgoszcz University of Science and Technology, 85-796 Bydgoszcz, Poland

**Keywords:** acute lymphoblastic leukemia classification, image background informativeness, Shannon entropy, cross-entropy, XGBoost

## Abstract

The study of leukemia classification using deep learning techniques has been conducted by multiple research teams worldwide. Although deep convolutional neural networks achieved high quality of sick vs. healthy patient discrimination, their inherent lack of human interpretability of the decision-making process hinders the adoption of deep learning techniques in medicine. Research involving deep learning proved that distinguishing between healthy and sick patients using microscopic images of lymphocytes is possible. However, it could not provide information on the intermediate steps in the diagnosis process. As a result, despite numerous examinations, it is still unclear whether the lymphocyte is the only object in the microscopic picture containing leukemia-related information or if the leukocyte’s surroundings also contain the desired information. In this work, entropy measures and machine learning models were applied to study the informativeness of both whole images and lymphocytes’ surroundings alone for Leukemia classification. This work aims to provide human-interpretable features marking the probability of sickness occurrence. The research stated that the hue distribution of images with lymphocytes obfuscated alone is informative enough to facilitate 93.0% accuracy in healthy vs. sick classification. The research was conducted on the ALL-IDB2 dataset.

## 1. Introduction

Acute lymphoblastic lekemia (ALL) diagnosis is closely associated with morphological changes in white blood cells (WBC, or leukocytes). ALL, known in the group of blood diseases, is characterized by the overproduction and continuous proliferation of malignant and immature white blood cells (referred to as lymphoblasts or blasts). Although the number of leukocytes can often be considered an essential indicator of pathological changes in the morphological picture of the blood, it is not always sufficient.

The detection of ALL and its subtypes is often accomplished by examining blood or bone marrow smears. According to the French-American-British (FAB) classification standard, ALL is classified into the L1, L2, and L3 subtypes. Assignment to the correct subtype is carried out according to observation of the nucleus’s morphology, including the affected cell’s pattern and variation in its shape. This procedure is generally accepted and known from numerous works of authors researching the detection and classification of leukocytes.

In clinical practice, microscopic examination of blood smears to verify ALL is based primarily on counting different types of white blood cells. Equally important is the analysis of the nuclear features of leukocytes, which are often distinguished by their pastel blue coloration. Nevertheless, their further evaluation becomes complicated, as according to the FAB, features such as the size, shape variation, and texture should be considered. After all, leukocytes can be distinguished by their size, color characteristics, the ratio of the nuclei to the cytoplasm contained in them, etc.

Due to the morphological diversity of white blood cells, the classification into ALL subtypes may not always be realized correctly. The reason for this can perhaps be found in the complexity of the backgrounds of microscopic images. One of the essential aspects is played by the surroundings of white blood cells, which occur in the setting of other morphotic components of the blood such as red blood cells or platelets. However, according to FAB classification, the background does not add significant information from the perspective of classification, although it can significantly hinder the correct classification of ALL. Our work measures background informativeness to determine the influence of the lymphocytes’ surroundings on correct Leukemia classification.

In their article [1], Andrade et al. performed an extensive evaluation of the leukocyte segmentation techniques by artificial intelligence systems developed by well-established research teams. The authors performed experiments on five databases: ALL-IDB2 [2], BloodSeg [3], Leukocytes [4], JTSC Database, and CellaVision [5]. The leukocyte segmentation methods examined by the authors use Otsu threshold [6,7,8,9] (Madhloom et al., Arslan et al., Nazlibilek et al., and Prinyakupt et al.), K-means [10,11,12,13,14,15,16,17] (Nasir et al., Mohapatra et al., Madhukar et al., Amin et al., Sarrafzadeh et al., Vincent et al., Vogado et al., and Kumar et al.), region growing [7,10,18] (Nasir et al., Mohammed et al., and Arslan et al.), edge detector [18] (Mohammed et al.), Zack’s algorithm [19] (Abdeldaim et al.), and arithmetical image processing operations [3,6,9,14] (Madhloom et al., Mohamed et al., Prinyakupt et al., and Sarrafzadeh et al.). The methods were examined on the images encoded using RGB [9] (Prinyakupt et al.), grayscale [6,18] (Madhloom et al., and Mohammed et al.), HSI [10] (Nasir et al.), L*a*b* [11,12,14,15] (Mohapatra et al., Madhukar et al., Sarrafzadeh et al., and Vincent et al.), HSV [13,17] (Amin et al., and Kumar et al.), CMYK [19] (Abdeldaim et al.), and CMYK + L*a*b* color schemes [16] (Vogado et al.). The authors achieved satisfactory results for all of the datasets, in some cases reaching 97% accuracy. However, none of the methods examined proved to be the best on all datasets. It is also important to note that the method with the best results in this experiment achieved only a 58.44% leukocyte nuclei detection rate. In this article, a leukocyte was considered detected by computing the true positive rate metric $TPR_{t}$ with a threshold t=0.9.

In [1], Andrade et al. proved that leukocyte nuclei segmentation is a non-trivial task, and none of the well-established methods proved efficient in leukocyte detection. The authors stated in the article that leukocyte nuclei segmentation is performed to classify the presence of leukemia in the cells. The survey of image processing techniques and their results motivated us to perform research focusing on attempting to examine the amount of information contained in non-leukocytes for leukemia classification without performing image segmentation. Furthermore, the plethora of black box-type artificial intelligence systems applied with various evaluation results signified the importance of establishing the features’ quality.

The subject of leukemia classification was researched thoroughly using deep learning methods. The authors of [20,21,22,23,24] applied various types of convolutional neural networks (CNNs) [25]. In [20], Rehman et al. applied the AlexNet [26] architecture of CNN networks for leukemia classification, achieving 97.78% accuracy. Similar research was conducted by the authors of [21] (Prellberg et al.) using the ResNeXt50 [27] architecture of CNN networks pretrained on the ImageNet dataset [28]. The researchers achieved 89.7% accuracy. The non-binary classification of leukemia was conducted by Ahmed et al. [22]. The experiments were run to establish the ability of CNNs to discriminate in one vs. many mode against leukemia types such as acute myeloid leukemia (AML), chronic lymphocytic leukemia (CLL), chronic myeloid leukemia (CLM), and acute lymphoblastic leukemia (ALL). They achieved classification with 81% accuracy. The authors of [23] (Guo et al.) used Siamese networks [29] to achieve few-shot learning [30] with 89.96% accuracy. Similar research was conducted by Abhishek et al. [24]. In this work, the authors used transfer learning to compare the results of deep convolutional neural networks with support vector machines (SVMs) against SVM interpreting features extracted by local binary patterns (LBPs) [31] and the histogram of oriented gradients (HOG) [32]. The deep learning approach obtained 98% accuracy, SVM + LBP resulted in 83% accuracy, and SVM + HOG resulted in 50% accuracy. In their work, Rodrigues et al. [33] also applied deep learning for leukemia classification. What differs them from other works is the optimization of trained neural networks using a genetic algorithm. This procedure improved the results to 98.46%.

The results of leukemia classification using deep learning in [20,21,22,23] (Rehman et al., Prellberg et al., Ahmed et al., Guo et al., Abhishek et al., and Rodrigues et al.) provided substantial accuracy for healthy vs. sick discrimination. However, due to the black box nature of deep learning methods, it is unclear what pattern was extracted by neural networks to achieve such efficiency. The knowledge distillation [34] techniques are currently not advanced enough to determine the neural network’s reasoning, leading to classification in a manner humans can understand. Because of this issue, the adoption of deep learning and machine learning in medicine is slow due to the inability to verify the quality of the extracted features. Our work examines the ALL-IDB dataset samples to determine the features understandable by humans, allowing reliable classification and, at the same time, determining the usefulness of the leukocyte’s surroundings in leukemia classification.

### 1.1. Summary of Surveyed Research Works

The surveyed works are summarized in Table 1 to present various method and image encoding technique combinations applied in a readable format.

### 1.2. The Aim of This Work

This work is a continuation of the research described in [35] (Pałczyński et al.). This article aims to establish the informativeness of a lymphocyte’s surroundings for leukemia classification. The classification is conducted on the features extracted from images with lymphocytes obfuscated using black rectangles. The classification results without information regarding lymphocytes are compared against the quality of discrimination in the unmodified images. The features extracted from the image are deterministic, human-interpretable qualities. The discrimination is performed using both simple, divergence-based clusterization (mean squared error and cross-entropy [36]) and by applying machine learning algorithms such as logistic regression [37] and the XGBoost algorithm [38]. The images used in this research are encoded using RGB and HSV methods.

This work aims to quantify the amount of information stored in human-interpretable features computed from an image with the lymphocytes obfuscated. This work aims not to achieve the best classification results but to determine how well the classification can be performed using a limited amount of information while remaining interpretable by humans. In our previous article, deep neural networks were applied to the raw images to perform the classification. Although the results were satisfactory, the inherent black box nature of deep neural networks prevented us from acquiring human-interpretable knowledge on the nature of this particular classification problem. This work aims to provide such information.

### 1.3. Summary of Our Contributions

Our main contributions can be summarized as follows:We examined the influence of lymphocyte obfuscation on acute lymphoblastic leukemia classification to evaluate its surroundings’ informativeness. The hue distribution of lymphocytes’ surroundings processed by the XGBoost algorithm resulted in classification with 93% accuracy.We evaluated the informativeness of channels’ value distributions of both the RGB and HSV color encodings. We determined that the channel encoding color green contained the most information, with an XGBoost classification accuracy of 96%. The same evaluation of red and blue color channels resulted in classification accuracies of 87% and 83%, respectively. The hue, saturation, and value channels obtained classification results of 94%, 94%, and 84%, respectively.The classification results of the XGBoost algorithm interpreting the distributions of individual channel values resulted in a classification quality similar to the effects of deep learning application on raw images performed by other researchers. As a result, we reduced the amount of input information by three orders of magnitude while achieving comparable results.We evaluated the informativeness of the entropy measurements of each channel’s values distribution using the Shannon entropy. The Shannon entropy computed for the hue distribution of images with lymphocytes obfuscated resulted in a classification accuracy of 81% and 68% accuracy when using images without the lymphocytes being obfuscated. The results suggest that lymphocytes’ surroundings contain essential information for acute lymphoblastic leukemia classification.

### 1.4. Paper Organization

This work is divided into sections. Section 2 describes the materials and methods used in this research. This section describes the image preprocessing techniques, encoding, feature vectorization, experiments conducted, and metrics. Section 3 presents the results of the experiments described in Section 2. Section 4 provides interpretation of the results presented in Section 3, and Section 5 concludes the paper.

## 2. Materials and Methods

This section describes the materials and methods used in this research. It describes the data preprocessing, vectorization, and algorithms for generating the experimental results. The experimental procedure involved computing the Shannon entropy, measuring the cross-entropy score between the obtained value distributions, and fitting the machine learning models.

### 2.1. ALL-IDB Database

The research was conducted on the ALL-IDB database, containing microscopic images of lymphocyte cells documented from healthy people and patients with acute lymphoblastic leukemia. This database is made publicly available by Universitià degli Studi di Milano, and it contains annotation of which samples represent cases of leukemia and which were obtained from healthy patients. Oncologists performed the annotation.

The database contains 260 microscopic images. The dataset is balanced between classes, having 130 images of blood smears taken from healthy patients and the same amount from sick ones. The home website of this dataset is accessible at this link: http://homes.di.unimi.it/scotti/all/ (accessed on 15 October 2021).

### 2.2. Image Preprocessing

In this section, the techniques of image processing applied during the experiments are presented. The amount of information contained within the background of the image can be examined by removing the lymphocytes from the graphics. Removing lymphocyte information was performed by covering them with black rectangle-shaped bounding boxes. This type of obfuscation was chosen to remove all information from the lymphocytes and information regarding the cells’ shapes, which may have interfered with the experiment’s results. The resulting shrinkage of the background from such an obfuscation technique was not considered a concern. The results of this operation are presented on the Figure 1. The experiments were conducted using images both unmodified and obfuscated to compare the information stored in background of the image with all of the information contained.

The images were also subjected to data augmentation to determine the influence of commonly used image preprocessing techniques on the informativeness of the background. The data augmentation was performed before lymphocyte obfuscation. The modification methods examined in this research were the following:Gaussian blur;Median blur;Gaussian noise.

Gaussian blur (also known as Gaussian smoothing) is a commonly used data augmentation technique in deep learning for increasing the number of images in the training set. It serves as a low-pass filter, reducing higher frequencies from the image and thus achieving the perceived effect of smoothness. The filter works by convolving the image with a matrix representation of a two-dimensional Gaussian function. Equation (Equation 1) presents the method for obtaining the filter matrix: (1)$$G\left(x,y\right)=\frac{1}{2\pi \sigma^{2}}e^{-\frac{x^{2}+y^{2}}{2\sigma^{2}}}$$
where *x* and *y* represent the divergence from the center of the matrix. The bigger the matrix (kernel), the more precise the convolution is. The kernel lengths used in this research were 3, 9, 21, and 51. The results of applying this operation are presented in Figure 2.

Median blur has a similar purpose to Gaussian blur. This filter windows the image and returns the median value from each window. This operation reduces noise and creates a low-pass filter. The sizes used for the windows in this research were 3, 9, 21, and 51. The results of applying this operation are presented in Figure 3.

The last data augmentation technique used was adding Gaussian noise. Compared with the previous two techniques, this method increases the amount of noise instead of decreasing it. It is also commonly used in deep learning to reduce deep neural networks’ overreliance on temporal patterns in favor of a more holistic approach. Each pixel in the output file was computed using Equation (Equation 2): (2)$$x^{\prime }=\left\lfloor x\cdot \left(1+n\right)\right\rfloor ,\;n∼N\left(0,\sigma \right)$$

Here, *x* is the current value of the filtered image, and σ is the standard deviation of the distribution. The values of variance used in this research were 0.001, 0.01, and 0.1. The results of applying this technique are presented in Figure 4. The processing data pipeline involving data augmentation and lymphocyte obfuscation is presented in Figure 5.

### 2.3. Image Vectorization

In this section, the preprocessed images are converted into vectors of features ready to be interpreted by the statistical and machine learning models. In this research, background informativeness was measured in regard to one image channel at a time. As a result, each experiment conducted started with selection of the aforementioned channel. The available channels were red (R), green (G), blue (B), hue (H), saturation (S), and value (V).

The first three channels (red, green, and blue) are natural components of RGB-encoded images. However, the hue, saturation, and value metrics are the result of HSV image encoding. HSV is a common type of color expression in an image more akin to the recognition process performed by the human eye.

The hue channel contains information on what color is present in the image. It represents a 360° coordinate of rotation around the circle of colors. Typical representation of the hue coordinates associate the color red with a value of 0°, yellow with 60°, green with 120°, aqua with 180°, blue with 240°, and purple with 300°. It is important to note that the hue represents a rotation angle, so the difference between two hue values is represented by the measurement of the shortest arc connecting two points on the hue circle. For example, the colors red (0°) and purple (300°) are 60° degrees apart instead of 300°. In this research, the OpenCV library was used for image processing, which encoded the hue channels with values from 0 to 180. This behavior was kept for both conducting experiments and presenting the results.

The saturation contains information regarding the color’s intensity. A saturation value of 0 results in the color gray, and the maximum value of 255 provides the most intense version of the color. On the other hand, the value channel contains information on the brightness of the color. The numeric value of 0 results in the color black, and the maximum value of 255 generates the brightest version of the color.

Equations (Equation 3)–(Equation 6) describe the process of image conversion from RGB encoding to HSV encoding:(3)M=max{R,G,B},m=min{R,G,B},
(4)V=M/255
(5)$$S=\left\{\begin{array}{cc} \frac{1-m}{M} & M>0\\ 0 & M=0 \end{array}\right.$$
(6)$$H=\left\{\begin{array}{cc} cos^{-1}\left(\frac{R-\frac{1}{2}G-\frac{1}{2}B}{\sqrt{R^{2}+B^{2}+G^{2}-RB-RG-GB}}\right) & G\geq B\\ 360^{{}^{\circ}}-cos^{-1}\frac{R-\frac{1}{2}G-\frac{1}{2}B}{\sqrt{R^{2}+B^{2}+G^{2}-RB-RG-GB}} & B>G \end{array}\right.$$

In the next step, the selected channel is vectorized by grouping its individual values into 30 equally spaced numeric bins and computing their distribution in the whole image. Such encoding provides information on what value occurs most frequently in the image. Pixels encoding black rectangles for obfuscation purposes were not included in the computations of color density distribution. An example of such calculations is presented in Figure 6.

### 2.4. Distribution Difference Measurement

In this section, images vectorized into a single channel’s values density distributions are compared to measure the amount of information stored in every channel’s histogram. Distributions from both the obfuscated and unmodified images are examined to establish the amount of information stored in the color distributions of both the entire images and the backgrounds only. The quality of image classification measures the information stored in color distributions regarding the representation of healthy and sick individuals.

In the first step of the experiment, the set of vectorized images and the corresponding set of labels are randomly split into training and test sets. Then, the distribution vectors from the training set were split into subsets representing each class. Next, the averaged distributions were computed from the subsets, and each computed average distribution represented the class from which the samples were computed. In another step, the divergence metric was chosen to measure how different the samples were from each representative distribution. The divergence metrics chosen in this research were the cross-entropy and mean squared error, and they are explained in detail in Section 2.4.1 and Section 2.4.2. The divergence metric was used to compute the divergence of each sample from the training set to each representative distribution, and their values were stored in the respective sets. Next, from each class divergence set, the mean and standard deviation were computed. Then, the statistical model training was finished and ready for performing test evaluation.

During the evaluation, the samples from the test set were subjected to divergence computation for the averaged distributions representing each class. The similarities to each distribution were normalized by subtracting the corresponding mean and dividing by the standard deviation. The normalized similarities of each sample to each distribution were compared. The sample was associated with the class whose representative distribution had the smallest normalized divergence. The evaluation of classification quality measures the amount of information contained in the value distribution. The process is graphically presented in Figure 7 and described in detail in the pseudo-code in Section 2.4.3.

#### 2.4.1. Cross-Entropy

Cross-entropy is a technique from information theory for computing the divergence between two probability distributions. Equation (Equation 7) describes the process of metric calculation. The value density distribution can be interpreted as the probability that an individual pixel has a certain value. Such an interpretation allows the usage of cross-entropy in value density distribution classification:(7)$$H\left(P,Q\right)=-\sum_{i=0}^{\left|P\right|}P_{i}\log Q_{i},|P|=\left|Q\right|$$
where *P* and *Q* are the density distributions subjected to the comparison.

#### 2.4.2. Mean Sqaured Error

The mean squared error (*MSE*) is a commonly used technique for comparing two vectors. It is described by Equation (Equation 8). Due to powering the differences, the *MSE* is prone to disregarding multiple closely matched dimensions of two vectors in favor of penalizing a few outlying ones. This feature is useful in vector comparison because it forces the algorithms to even out their match functions instead of attuning easily matchable parts of the vector and disregarding the difficult ones: (8)$$MSE\left(P,Q\right)=\sum_{i=0}^{\left|P\right|}{\left(P_{i}-Q_{i}\right)}^{2},|P|=\left|Q\right|$$
where *P* and *Q* are the vectors subjected to the comparison.

#### 2.4.3. Algorithm

The graphical representation of the experiment process is depicted in Figure 7. The pseudo-code describing this process in detail is presented below Algorithm 1:
**Algorithm 1:** The mathematical formulation of experimental procedure examining distribution difference measurementsInput the set of samples *X*;Input the set of sample labels *Y*;Input divergence function fn;Randomly shuffle the set of indices $I=\{i|i\in N^{+}\cap i\in \left(0;|X\right|+1)\}$;$s=\lfloor \frac{2}{3}|X|\rfloor$;$I_{train}=I\left[:s\right]$;$I_{test}=I\left[s:\right]$;$X_{train}=\left\{X_{i}|i\in I_{train}\right\}$;$Y_{train}=\left\{Y_{i}|i\in I_{train}\right\}$;$X_{test}=\left\{X_{i}|i\in I_{test}\right\}$;$Y_{test}=\left\{Y_{i}|i\in I_{test}\right\}$;$X_{0}=[\{X_{train}\left[i\right]|i\in (0,|X_{train}|+1)\cap Y_{train}\left[i\right]=0\}]$;$X_{1}=[\{X_{train}\left[i\right]|i\in (0,|X_{train}|+1)\cap Y_{train}\left[i\right]=1\}]$;$M_{0}=[\{\frac{1}{|X_{0}|}\sum_{j\in X_{0}}X_{0}\left[j,i\right]|i\in N^{+}\cap i\in |X_{0}\left[0\right]|\}]$;$M_{1}=[\{\frac{1}{|X_{1}|}\sum_{j\in X_{1}}X_{1}\left[j,i\right]|i\in N^{+}\cap i\in |X_{1}\left[0\right]|\}]$;$D_{0}=\{fn\left(X_{train}\left[i\right],M_{0}\right)|i\in N^{+}\cap i\in \left(0;\right|X_{train}\left|+1\right\}$;$D_{1}=\{fn\left(X_{train}\left[i\right],M_{1}\right)|i\in N^{+}\cap i\in \left(0;\right|X_{train}\left|+1\right\}$;$m_{0},s_{0},m_{1},s_{1}=mean\left(D_{0}\right),std\left(D_{0}\right),mean\left(D_{1}\right),std\left(D_{1}\right)$;$Dt_{0}=\{\frac{1}{s_{0}}\left(fn\left(X_{test}\left[i\right],M_{0}\right)-m_{0}\right)|i\in N^{+}\cap i\in \left(0,\right|X_{test}\left|+1)\right\}$;$Dt_{1}=\{\frac{1}{s_{1}}\left(fn\left(X_{test}\left[i\right],M_{1}\right)-m_{1}\right)|i\in N^{+}\cap i\in \left(0,\right|X_{test}\left|+1)\right\}$;$P=[\{Dt_{1}\left[i\right]<Dt_{0}\left[i\right]|i\in N^{+}\cap i\in (0;|Dt_{0}\left|+1)\}\right]$;Compare the prediction vector *P* with the label vector $Y_{test}$.

### 2.5. Shannon Entropy

The Shannon entropy is a mathematical tool from the field of information theory that allows measuring the amount of uncertainty the probability distribution contains. The more evenly spaced the probability among the distribution states, the higher the value of the Shannon entropy. The computation of the Shannon entropy is performed using Equation (Equation 9): (9)$$H\left(P\right)=-\sum_{i=0}^{\left|P\right|}P_{i}\log P_{i}$$
where *P* is the vectorized density distribution subjected to the Shannon entropy computation.

In this research, the Shannon entropy was used to quantify the uncertainty associated with each channel’s value distribution and evaluate whether there was a significant difference between the Shannon entropy of the samples from the healthy class and the entropy of the sick class of samples. The significance of the difference in entropy measurements was established by using the Shannon entropy as a single-value determinant in the classification of whether a patient was healthy or sick. The classification was performed by fitting the logistic regression model on randomly split training data and evaluating it on the test data.

### 2.6. Machine Learning Algorithms

The last experiment aimed to apply machine learning algorithms directly to the channel’s value distribution to attempt to perform the classification. The algorithms used for this task were XGBoost and logistic regression. The former is one of the most robust, state-of-the-art machine learning algorithms capable of extracting complex, multi-dimensional patterns. The latter is one of the simplest classification algorithms, providing a basis for comparison.

### 2.7. Metrics

The metrics used for classification quality measurement were *accuracy*, *precision*, *recall*, and $F_{1}$ score. The metrics are described by Equations (Equation 10)–(Equation 13). The following abbreviations are used to simplify these equations:TP = true positive;TN = true negative;FP = false positive;FN = false negative.
(10)$$Accuracy=\frac{TP+TN}{TP+FP+TN+FN}$$
(11)$$Precision=\frac{TP}{TP+FP}$$
(12)$$Recall=\frac{TP}{TP+FN}$$
(13)$$F_{1}=\frac{2TP}{2TP+FN+FP}$$

The accuracy is a global evaluation metric, and it assesses the model compared to all the data. On the other hand, the precision, recall, and $F_{1}$ score are local metrics that evaluate performance regarding the classification of one class vs. all of them.

## 3. Results

The results of the experiments described in Section 2.4, Section 2.5 and Section 2.6 are presented in Table 2, Table 3, Table 4, Table 5, Table 6, Table 7 and Table 8. Each experiment was repeated 30 times with randomly selected training and test sets. The tables present the averaged metric values from 30 trials. Section 3.1 presents the values acquired from running experiments on images with and without obfuscation without adding data augmentation. Section 3.3 examines the influence of data augmentation. The experiments were run on the images with and without obfuscation, and data augmentation was applied. The results were presented for the channel, and the experimental results presented in Section 3.1 contained the most information.

### 3.1. Background Information Measurement

This subsection presents the information measurements in the image’s background in the form of the classification quality and compares it against the information in the whole image. Table 2 contains an evaluation of the distribution difference measurement described in Section 2.4. Table 3 presents the results from evaluating the Shannon entropy of the value distribution as a sole healthy or sick determinant. The experiment procedure is described in Section 2.5. Table 4 contains information on the machine learning algorithm’s performance for the value distribution.

Each table contains the column “Lymphocytes Obfuscated”. The rows with the value “True” in this column contain results from experiments with parts of the image representing lymphocytes covered by a black rectangle. As a result, these rows depict the informativeness of the image background. The ones with “False” in the first column contain results from experiments that used whole images and provide the informativeness measurements of all images. These serve as a basis of comparison for the experiments with obfuscated lymphocytes.

Each table contains the column “Channel” as well. This column presents the information regarding channel selection for the experiment. Each experiment evaluated the informativeness of only one channel at a time to establish whether the background in any channel contained unwanted information.

Figure 8 represents the averaged value distribution obtained from each channel for each of four states: images of healthy patients without lymphocytes obfuscated, images of healthy patients with lymphocytes obfuscated, images of sick patients without lymphocytes obfuscated, and images of sick patients with lymphocytes obfuscated. These four states are represented in their respective columns. The rows of the chart grid represent each of the six channels: red, green, blue, hue, saturation, and value.

### 3.2. Comparison with the Literature

A comparison of the most promising models obtained in this research with other works is presented in Table 5. The results were compared against the outcomes of our previous work and work of Rodrigues et al. [33,35], which according to our literature review obtained the best results on the ALL-IDB2 dataset.

### 3.3. Influence of Data Augmentation

Table 2, Table 3 and Table 4 prove that the channel containing the most informative image background was the hue channel, with the classification accuracy obtained from merely a background hue value distribution ranging from 82% to 93%. As a result, this channel was subjected to further investigation to determine the data augmentation application’s influence on obfuscated and unmodified images.

Table 6 contains the results of applying Gaussian blur with kernel sizes of 3, 9, 21, and 51. Table 7 presents the results of median blur application with kernel sizes of 3, 9, 21, and 51. Table 8 presents influence of Gaussian noise application on the images with variances of 0.0001, 0.01, and 0.1. The experiments were conducted with and without lymphocyte obfuscation. Each experiment was repeated 30 times, and its values were averaged.

## 4. Discussion

The experimental results presented in Table 2, Table 3 and Table 4 determined that the hue channel contained the highest amount of image background information. All methods (except for logistic regression) achieved averaged test accuracies above 80%, with the XGBoost model having 93% accuracy. Such scores were obtained merely for the hue distribution of the image background, with the informativeness unconfirmed by academic knowledge.

The methods for background information measurement described in Section 2.4, Section 2.5 and Section 2.6 proved to be efficient in determining whether the background contained classification-sensitive information. These methods can be used as training dataset evaluation techniques. Suppose that a supposedly neutral classification-wise background contains the required information. In this case, artificial intelligence models such as deep convolutional neural networks may learn to recognize some unwanted, dataset-related temporal pattern in the background instead of true generalization to real-world scenarios. This method can help in the evaluation of datasets containing high-quality data.

The logistic regression model trained on raw distributions achieved the worst accuracy, and the XGBoost model achieved the best accuracy. This suggests that, although the information contained in the distribution of the values is substantial, it is not yet obvious. Such background information may not be caught during exploratory data analysis and interfere with machine learning algorithms’ training quality. For these reasons, the background informativeness evaluation may prove beneficial in fool-proofing artificial intelligence systems.

Data augmentation techniques reduced the background informativeness extracted by the application of cross-entropy. It had little effect on the Shannon entropy-based classification and did not affect the XGBoost or *MSE* classification quality. This suggests that information in the background of ALL-IDB images may be more complex than just random class-specific noise. The authors plan to investigate this phenomenon further.

The research indicates that information is contained in the hue distribution of ALL-IDB image backgrounds. The XGBoost model achieved 93% accuracy on merely the hue distribution in the background. Such high classification quality has been achieved by just studying the background, which is supposed to be classification-neutral. According to our literature review, the primary indication of acute lymphoblastic leukemia is an examination of the lymphocytes. Academics do not unanimously recognize the lymphocytes’ surroundings’ informativeness. However, this research proves that this is not the case in this dataset. It is possible that the suspected “classification-neutral background” contains information allowing for healthy and sick discrimination. The authors plan to investigate this phenomenon further.

In our previous work [35], the best combination of artificial neural networks for raw image encoding, classification heads, and image augmentation resulted in an average classification quality of 94.8%. The neural network used for this task was MobileNet v2, the state-of-the-art neural network for numerous image-processing tasks containing 3.4 million parameters. In this work, the XGBoost algorithm alone, which interpreted the green color value distribution, achieved a classification accuracy of 96.0%. A similar result was obtained by the XGBoost algorithm interpreting the hue distributions of images with lymphocytes obfuscated, achieving a classification quality of 93.0%. Much simpler machine learning models operating on limited data obtained results comparable to the state-of-the-art deep learning method. This suggests that the neural networks experienced overfitting during training despite the application of data augmentation techniques. This also suggests that the task of leukemia classification may be performed using much more straightforward and cost-effective methods that also benefit from human interpretability. The authors plan to investigate this phenomenon further. According to our literature review, at the moment of writing this article, the best result was obtained by Rodrigues et al. [33], with an accuracy of 98.5%. This result is 2.5% percentage points higher than our best model. However, we obtained our results using around 3000 times fewer input data and almost 4000 times fewer parameters. As a result, our best model obtained comparable results, requiring much less computational power while remaining interpretable by humans.

The results indicate that hue distribution of a lymphocyte’s surroundings contains information supporting leukemia classification. However, a distribution is, by definition, an aggregation of the information stripped from temporal patterns akin to hidden Markov chains. It is possible that more detailed information interpretable by humans can be found during image examination in the hue channel. The authors plan to investigate this claim further.

## 5. Conclusions

The proposed background informativeness measurement proved its efficiency in dataset quality evaluation. This method based on the Shannon entropy, cross-entropy, and machine learning algorithms provides a comprehensive estimation of value distribution patterns in the background that may cause artificial intelligence models to overfit them instead of finding generalized solutions applicable to real-world problems.

The conducted research on background informativeness on the ALL-IDB dataset found a substantial amount of information in the hue distribution of the image background. The hue distributions of healthy people and patients who had acute lymphoblastic leukemia differed vastly from each other and by their Shannon entropy measurements. In this research, the lymphocytes were obfuscated with a black rectangle, so this information was contained within the supposedly classification-neutral background. The authors plan to investigate this phenomenon further.

The highest quality of classification was achieved while examining the green channel distribution using the XGBoost model. On average, it achieved 96.0% accuracy. This is a result comparable with deep neural networks while requiring much less computational power and providing a more human-interpretable decision process.

The background hue distribution differences between the images of healthy and sick patients require further investigation. It is unknown whether the differentiable factor is spread uniformly over the whole picture or is concentrated around semantically separable entities. Medical professionals must examine the nature of these changes to understand the features’ origins and extrapolate the applicability of this knowledge. The authors plan to investigate this phenomenon further.

## Figures and Tables

**Figure 1 entropy-24-01560-f001:**
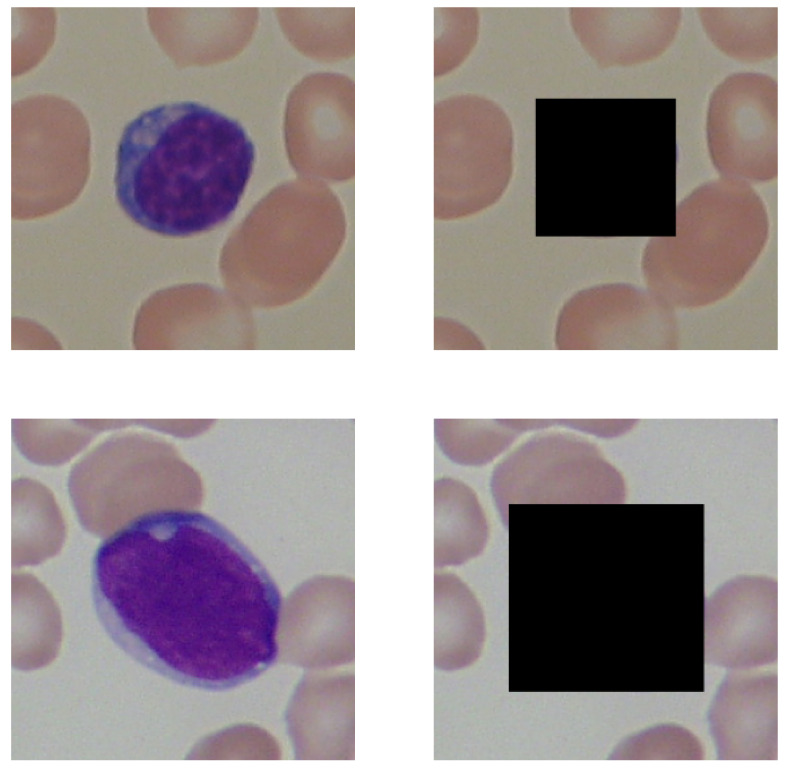
A black rectangle covers an example of lymphocytes in the images in the second column. The first row presents versions of the image of a healthy patient, and the second row depicts versions of the photo taken for a patient sick with leukemia.

**Figure 2 entropy-24-01560-f002:**

Presentation of the Gaussian blurring application on the images. The blurring procedure did not affect the first image in the row, and the rest were convolved with kernel of sizes 3, 9, 21, and 51 pixels, respectively.

**Figure 3 entropy-24-01560-f003:**

Presentation of the median blurring application on the images. The blurring procedure did not affect the first image in the row, and the rest were convolved with kernel of sizes 3, 9, 21, and 51 pixels, respectively. The images were 257 × 257 pixels in size.

**Figure 4 entropy-24-01560-f004:**
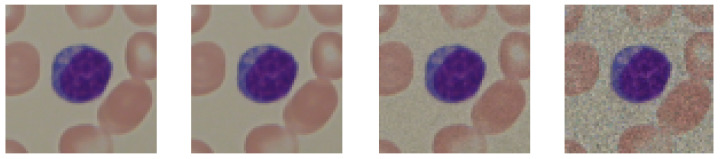
Presentation of the Gaussian noise application on the images. The noise procedure did not affect the first image in the row, and the rest were subjected to the multiplicative noise sampled from the Gaussian distribution, parametrized by the mean equal to zero and variance containing values of 0.001, 0.01, and 0.1, respectively.

**Figure 5 entropy-24-01560-f005:**
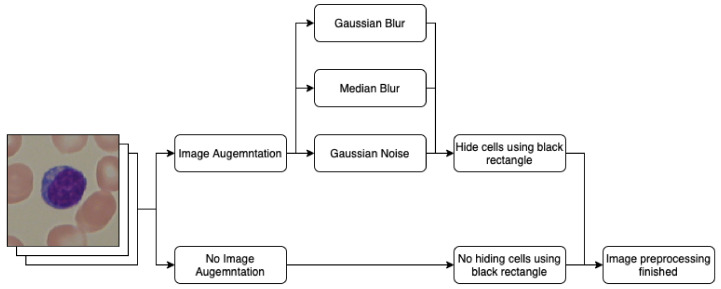
Graphical representation of image processing procedure.

**Figure 6 entropy-24-01560-f006:**
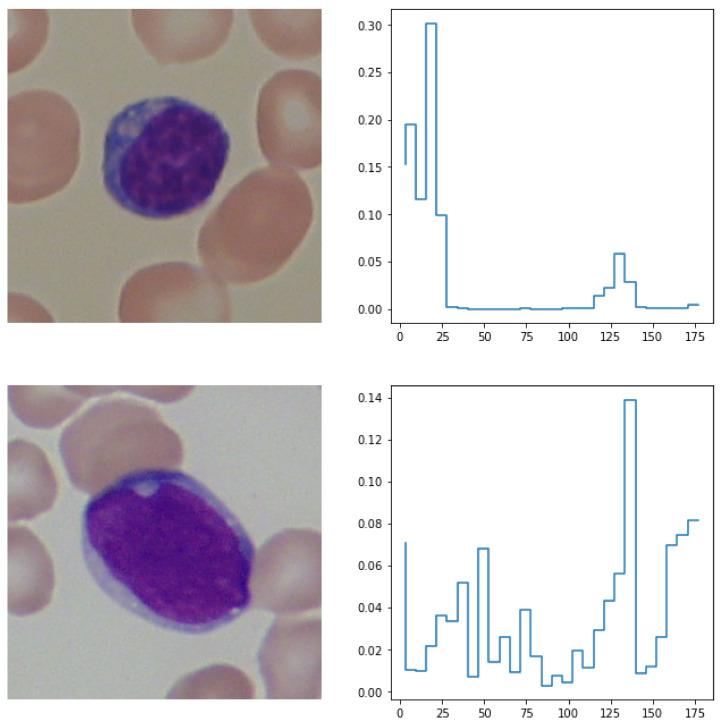
A presentation of images taken from healthy and sick patients (first column) and their corresponding hue distributions. HSV encoding was applied to both of the images.

**Figure 7 entropy-24-01560-f007:**
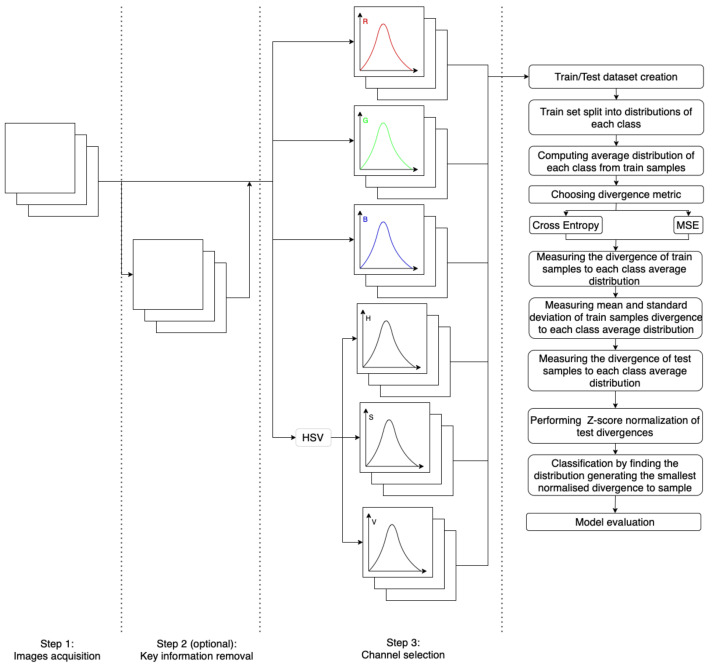
A graphical representation of the classification process involving images’ color distribution comparison, using cross-entropy and mean squared error (*MSE*) as divergence metrics.

**Figure 8 entropy-24-01560-f008:**
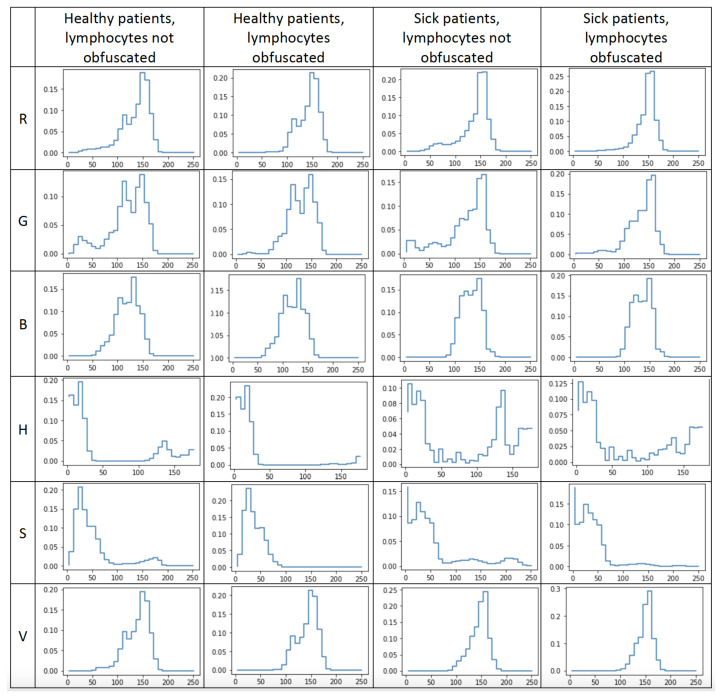
Presentation of averaged distributions of each channel’s values for each combination of images. The columns represent distributions taken from images of healthy patients without lymphocytes covered, images of healthy patients with lymphocytes covered, images of sick patients without lymphocytes covered, and images of sick patients with lymphocytes covered. Each row represents one channel. The presented channels are red (R), green (G), blue (B), hue (H), saturation (S), and value (V).

**Table 1 entropy-24-01560-t001:** A summary of investigated research works.

Ref.	Image Encoding	Methods
[3]	Grayscale	Arithmetical operations
[6]	Grayscale	Otsu threshold, arithmetical operations
[7]	RGB	Otsu threshold, region growing
[8]	Grayscale	Otsu threshold
[9]	RGB	Otsu threshold, arithmetical operations
[10]	HSI	K-means, region growing
[11]	L*a*b*	K-means
[12]	L*a*b*	K-means
[13]	HSV	K-means
[14]	L*a*b*	K-means, arithmetical operations
[15]	L*a*b*	K-means
[16]	CMYK + L*a*b*	K-means
[17]	HSV	K-means
[18]	Grayscale	Region growing, edge detectors
[19]	CMYK	Zack’s algorithm
[20]	RGB	AlexNet
[21]	RGB	ResNeXt50
[22]	RGB	CNN
[23]	RGB	Siamese networks
[24]	RGB	CNN
[33]	RGB	ResNet50 v2

**Table 2 entropy-24-01560-t002:** The classification results by the images’ color distribution comparison to the averaged distributions representing their respective classes. The cross-entropy and mean squared error (*MSE*) metrics were applied. The images were unmodified and had their lymphocytes covered (first column). No image augmentation was applied in this experiment.

Lymphocytes Obfuscated	Channel	Cross Entropy Acc.	Cross Entropy $F_{1}$ (Healthy)	Cross Entropy $F_{1}$ (Sick)	MSE Acc.	MSE $F_{1}$ (Healthy)	MSE $F_{1}$ (Sick)
False	B	0.50	0.56	0.41	0.68	0.68	0.68
True	B	0.55	0.59	0.49	0.69	0.68	0.70
False	G	0.81	0.81	0.81	0.62	0.63	0.61
True	G	0.47	0.53	0.38	0.60	0.62	0.58
False	R	0.53	0.51	0.54	0.46	0.49	0.43
True	R	0.37	0.49	0.18	0.45	0.48	0.42
False	Hue	0.85	0.86	0.83	0.77	0.79	0.73
True	Hue	0.83	0.85	0.81	0.82	0.84	0.80
False	Saturation	0.79	0.80	0.77	0.79	0.83	0.72
True	Saturation	0.73	0.72	0.74	0.79	0.83	0.73
False	Value	0.45	0.51	0.36	0.50	0.51	0.48
True	Value	0.38	0.45	0.28	0.52	0.53	0.49

**Table 3 entropy-24-01560-t003:** The classification results by the images’ color distributions’ Shannon entropy measurements. The images were unmodified and had their lymphocytes covered (first column). No image augmentation was applied in this experiment.

Lymphocytes Obfuscated	Channel	Avg. Shannon Entropy (Healthy)	Std. Shannon Entropy (Healthy)	Avg. Shannon Entropy (Sick)	Std. Shannon Entropy (Sick)	Acc	$F_{1}$ (Healthy)	$F_{1}$ (Sick)
False	B	0.43	0.01	0.45	0.02	0.51	0.31	0.51
True	B	0.43	0.02	0.44	0.02	0.51	0.30	0.51
False	G	0.48	0.02	0.50	0.02	0.55	0.37	0.55
True	G	0.43	0.03	0.46	0.03	0.54	0.39	0.54
False	R	0.44	0.02	0.47	0.02	0.54	0.38	0.54
True	R	0.39	0.03	0.42	0.04	0.54	0.39	0.54
False	Hue	0.63	0.03	0.71	0.07	0.68	0.70	0.65
True	Hue	0.57	0.03	0.69	0.10	0.81	0.84	0.78
False	Saturation	0.49	0.03	0.51	0.02	0.55	0.38	0.55
True	Saturation	0.44	0.03	0.47	0.04	0.51	0.36	0.51
False	Value	0.43	0.02	0.44	0.02	0.51	0.30	0.51
True	Value	0.39	0.03	0.41	0.03	0.52	0.34	0.53

**Table 4 entropy-24-01560-t004:** The classification results by images’ color distribution interpretation by machine learning algorithms. The XGBoost and logistic regression models were used in this experiment. The images were unmodified and had their lymphocytes covered (first column). No image augmentation was applied in this experiment.

Lymphocytes Obfuscated	Channel	XGBoost Acc	XGBoost $F_{1}$ (Healthy)	XGBoost $F_{1}$ (Sick)	Logistic Regression Acc.	Logistic Regression $F_{1}$ (Healthy)	Logistic Regression $F_{1}$ (Sick)
False	B	0.83	0.83	0.83	0.53	0.32	0.53
True	B	0.82	0.82	0.82	0.55	0.37	0.55
False	G	0.96	0.96	0.96	0.50	0.28	0.51
True	G	0.86	0.87	0.85	0.50	0.28	0.51
False	R	0.87	0.87	0.86	0.47	0.23	0.49
True	R	0.80	0.80	0.79	0.48	0.28	0.49
False	Hue	0.94	0.95	0.94	0.57	0.41	0.57
True	Hue	0.93	0.94	0.93	0.60	0.46	0.59
False	Saturation	0.94	0.94	0.94	0.54	0.32	0.54
True	Saturation	0.88	0.89	0.88	0.55	0.36	0.55
False	Value	0.84	0.84	0.84	0.48	0.25	0.49
True	Value	0.80	0.81	0.80	0.50	0.30	0.50

**Table 5 entropy-24-01560-t005:** Comparison of the best models with and without lymphocytes obfuscated against other works. Shortcut “dist” stands for “distribution”.

Article	Input Data	Input Size	Obfuscation	Parameters	Accuracy	Precision	Recall	$F_{1}$
This work	Green dist.	51	False	6.4 K	0.960	0.960	0.960	0.959
This work	Hue dist.	36	True	6.4 K	0.935	0.935	0.935	0.934
[35]	RGB image	150 K	False	3.4 M	0.948	0.950	0.951	0.948
[33]	RGB image	150 K	False	25 M	0.985	0.986	0.985	0.984

**Table 6 entropy-24-01560-t006:** Results of Gaussian blur application on the quality of classification.

Lymphocytes Obfuscated	Kernel	XGBoost Acc.	Logistic Regression Acc.	Cross-Entropy Acc.k	MSE Acc.	Shannon Acc.
False	51	0.93	0.60	0.79	0.74	0.64
False	21	0.93	0.58	0.79	0.77	0.69
False	0	0.94	0.57	0.85	0.77	0.68
False	9	0.95	0.58	0.83	0.78	0.69
False	3	0.95	0.57	0.83	0.77	0.69
True	51	0.91	0.65	0.82	0.83	0.79
True	21	0.93	0.63	0.78	0.82	0.82
True	9	0.93	0.62	0.83	0.81	0.82
True	3	0.93	0.61	0.82	0.82	0.81
True	0	0.93	0.60	0.83	0.82	0.81

**Table 7 entropy-24-01560-t007:** Results of median blur application on the quality of classification.

Lymphocytes Obfuscated	Kernel	XGBoost Acc.	Logistic Regression Acc.	Cross-Entropy Acc.k	MSE Acc.	Shannon Acc.
False	51	0.94	0.59	0.73	0.77	0.67
False	0	0.94	0.57	0.85	0.77	0.68
False	9	0.95	0.58	0.81	0.79	0.69
False	21	0.95	0.59	0.81	0.79	0.69
False	3	0.95	0.57	0.83	0.77	0.69
True	51	0.91	0.63	0.72	0.80	0.80
True	3	0.93	0.61	0.82	0.82	0.81
True	9	0.93	0.63	0.81	0.81	0.82
True	0	0.93	0.60	0.83	0.82	0.81
True	21	0.94	0.65	0.78	0.81	0.81

**Table 8 entropy-24-01560-t008:** Results of multiplicative Gaussian noise application on the quality of classification.

Lymphocytes Obfuscated	Kernel	XGBoost Acc.	Logistic Regression Acc.	Cross-Entropy Acc.k	MSE Acc.	Shannon Acc.
False	0	0.94	0.57	0.84	0.76	0.68
False	0.0001	0.95	0.57	0.84	0.76	0.68
False	0.01	0.95	0.56	0.84	0.76	0.68
False	0.1	0.95	0.56	0.82	0.76	0.68
True	0.01	0.93	0.59	0.82	0.82	0.80
True	0.0001	0.93	0.60	0.83	0.82	0.81
True	0	0.93	0.60	0.82	0.82	0.81
True	0.1	0.95	0.59	0.82	0.82	0.81

## Data Availability

The data presented in this study are available on request from the corresponding author.

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
