# Peer review of "Entropy Measurements for Leukocytes’ Surrounding Informativeness Evaluation for Acute Lymphoblastic Leukemia Classification"

_entropy, 2022, doi:10.3390/e24111560_

Round 1
Reviewer 1 Report
The work presented is very interesting and the subject is well thought out. However, there are some areas of vagueness:
(i) the introduction has the quality of being rather long and detailed. However, it is not clear in the synthesis how the authors of [x] did this, for the authors of [y]. Please put the real names of the authors in the text.
(ii) some parts of the introduction are confusing, with there being method X which gives R1 as the result on dataset Z1, and R2 on Z2, but in fact the best result is only Z3 (Z3 <<Z2). This is not clear.
(iii) what is the specific question asked for this work. It should be put back more synthetically and with less bias.
(iv) the methodology section is a bit long with parts that are too detailed when they are trivial.
(v) It would be good to have more detail on the data that is essential. The number seems to me to be small and especially from only one source. A possibility of bias. It would be necessary to characterise them between classes and within classes. Is this a very easy game in the end. What is the impact with images coming from another location?
(vi) the beginning of the results is in the Materials and Methods.
(vii) The big problem is that indeed the reader can read the tables and look at the Figures but when there is no analysis (of the text), it is often that the work is not very interesting. Here the results part at 6 Tables & 1 Figure, with half a page away from what's in the Table boxes and ... really nothing else. This is not possible in a scientific journal.
(viii) the discussion is very short. As there is no analysis of the problematic data, there is not much.
In conclusion, some methodological work, but not enough depth.
Author Response
Thank you very much for your comments. They provided much-needed feedback on how to improve our work, and we made our best effort to fix all mentioned shortcomings of this work.
Comments:
- the introduction has the quality of being rather long and detailed. However, it is not clear in the synthesis how the authors of [x] did this, for the authors of [y]. Please put the real names of the authors in the text.
- some parts of the introduction are confusing, with there being method X which gives R1 as the result on dataset Z1, and R2 on Z2, but in fact the best result is only Z3 (Z3 <<Z2). This is not clear.
- what is the specific question asked for this work. It should be put back more synthetically and with less bias.
- the methodology section is a bit long with parts that are too detailed when they are trivial.
- It would be good to have more detail on the data that is essential. The number seems to me to be small and especially from only one source. A possibility of bias. It would be necessary to characterise them between classes and within classes. Is this a very easy game in the end. What is the impact with images coming from another location?
- the beginning of the results is in the Materials and Methods.
- The big problem is that indeed the reader can read the tables and look at the Figures but when there is no analysis (of the text), it is often that the work is not very interesting. Here the results part at 6 Tables & 1 Figure, with half a page away from what's in the Table boxes and ... really nothing else. This is not possible in a scientific journal.
- the discussion is very short. As there is no analysis of the problematic data, there is not much.
Responses:
- Names of the authors were added
- The numerous articles describing works conducted on multiple datasets were cited to provide more detailed view on the current state-of-the-art methods and techniques. They weren’t mentioned to be compared against our work, but to signify the experimental niche our work can fill, which is absence of human-interpretable machine learning articles.
- The main goal of this work has been stated more clearly in the introduction. The goal: “This work aims to quantify the amount of information stored in human-interpretable features computed from the image with lymphocytes obfuscated. This work aims not to achieve the best classification results but to determine how well the classification can be performed using the limited amount of information while remaining human-interpretable. In our previous article, the Deep Neural Networks were applied to the raw images to perform the classification. Although the results were satisfactory, the inherent black-box nature of Deep Neural Networks prevented us from acquiring human-interpretable knowledge on the nature of this particular classification problem. This work aims to provide such information.”
- The part describing color channels in two-level list was removed
- As stated in the article, this work is a continuation of our article: “IoT application of transfer learning in hybrid artificial intelligence systems for acute lymphoblastic leukemia classification”. That particular work was conducted solely on the ALL-IDB 2 dataset. This article is a deeper analysis of patterns present in the dataset with an additional focus on the human interpretability of extracted features. We decided to perform research on the ALL-IDB 2 dataset to compare the results with those previously obtained and conclude whether the application of Deep Neural Networks containing millions of parameters is viable compared to much simpler models. The discussion was extended to signify the continuation of the research better.
- I am sorry, but I did not find where the results are described in the Materials and Methods section. Could you please point out where do you see the problem?
- We opted to analyze data in the Discussion section and provide raw experimental measurements in the Results section. Such text structure was expected from us when we published our previous works in this journal. We decided that such a distinction between presenting the results and interpreting them increases the clarity of the work. We also expanded the analysis of the results in the Discussion.
- The discussion was expanded.
Reviewer 2 Report
The manuscript is about the leukemia classification quality method using image processing with machine learning. This subject is a good fit in the journal scope.
I fount shortcomings:
1. The article is not well presented in the abstract. In the abstract, the authors should show (in an overall way) how better their method is.
2. Abbreviations should be explained on the first occurrence, even in the abstract.
3. Row 338, I expected "figure 10", not "6".
4. All figures should have been referenced before their place in the text.
5. Referenced articles are too old. There is only one item from the last three years. Therefore the background of research presented in the first chapter, based on too old articles, is incomplete. The authors should add recent articles to the references list, rewrite the research's background, and rewrite the conclusion. Machine learning is the fastest-growing field of computer science, and references older than three years are usually too old and outdated.
6. Axes of charts in figure 10 have no descriptions and dimensions.
7. In figure 10, there is no description of every chart. It is not clear which is which.
Author Response
Thank you for your feedback. It provided much-needed insight into our work from a different angle. We made our best to improve our work based on it.
Comments:
- The article is not well presented in the abstract. In the abstract, the authors should show (in an overall way) how better their method is.
- Abbreviations should be explained on the first occurrence, even in the abstract.
- Row 338, I expected "figure 10", not "6".
- All figures should have been referenced before their place in the text.
- Referenced articles are too old. There is only one item from the last three years. Therefore the background of research presented in the first chapter, based on too old articles, is incomplete. The authors should add recent articles to the references list, rewrite the research's background, and rewrite the conclusion. Machine learning is the fastest-growing field of computer science, and references older than three years are usually too old and outdated.
- Axes of charts in figure 10 have no descriptions and dimensions.
- In figure 10, there is no description of every chart. It is not clear which is which.
Responses:
- The abstract was improved.
- The abstract was corrected. We found no more unexplained abbreviations in the main text.
- It is corrected.
- Unfortunately, this is a work of LaTeX template. It decides where figures are put.
- More recent works have been added
- These as histograms of probability density. They have no units, so their axes are descriptive.
- The image was improved for better readability
Reviewer 3 Report
-
In this paper, three authors (specialized in Telecommunications, Telecommunications, and Mechanical Engineering), and none of them specialized in any healthcare/medicine track. While the paper relates to healthcare/medicine applications, the authors should consult a disciplined person in one of the healthcare/medicine tracks to ensure highly confident medical details and information. At least the introduction section and the results should be confirmed by specialists to ensure the usefulness of information and insights.
-
The structure of the abstract is not professional. The abstract should be re-written as: Introduction- research gap/motivation - problem statement - proposed solution/methodology - results and comparisons - conclusions.
- The summary of contributions is not provided explicitly. The authors need to provide a subsection to summarize the list of contributions of their current research.
- The Literature Review section is not well developed. I suggest adding a comparative table toward the end of ''Introduction '' to contrast your solution with other state-of-art-related works.
- Also, the model is poorly validated via only one dataset (i.e., Bot-IoT Dataset); I suggest the authors validate their models via some other dataset to provide a comparative study and strengthen the argument f this article.
- The future directions of the authors’ contribution are not provided. I suggest that the authors state some explicit future recommendations to improve or upgrade their model.
- The experimental results obtained from the proposed model evaluation seem not attractive. However, the authors must provide a comparison with other state-of-the-art models in the same area of study. The comparison should consider all relevant performance factors such as accuracy, prediction time, precision,..etc.
- The authors need to proofread the paper to avoid any typos. Make sure that all references are cited in order of their appearance.
Author Response
Thank you for your suggestions. Your insightful comments forced us to rethink our approach to the work. I hope you find it better than the last time you read it.
Comments:
- In this paper, three authors (specialized in Telecommunications, Telecommunications, and Mechanical Engineering), and none of them specialized in any healthcare/medicine track. While the paper relates to healthcare/medicine applications, the authors should consult a disciplined person in one of the healthcare/medicine tracks to ensure highly confident medical details and information. At least the introduction section and the results should be confirmed by specialists to ensure the usefulness of information and insights.
- The structure of the abstract is not professional. The abstract should be re-written as: Introduction- research gap/motivation - problem statement - proposed solution/methodology - results and comparisons - conclusions.
- The summary of contributions is not provided explicitly. The authors need to provide a subsection to summarize the list of contributions of their current research.
- The Literature Review section is not well developed. I suggest adding a comparative table toward the end of ''Introduction '' to contrast your solution with other state-of-art-related works.
- Also, the model is poorly validated via only one dataset (i.e., Bot-IoT Dataset); I suggest the authors validate their models via some other dataset to provide a comparative study and strengthen the argument f this article.
- The future directions of the authors’ contribution are not provided. I suggest that the authors state some explicit future recommendations to improve or upgrade their model.
- The experimental results obtained from the proposed model evaluation seem not attractive. However, the authors must provide a comparison with other state-of-the-art models in the same area of study. The comparison should consider all relevant performance factors such as accuracy, prediction time, precision,..etc.
- The authors need to proofread the paper to avoid any typos. Make sure that all references are cited in order of their appearance.
Responses:
- „At least the introduction section and the results should be confirmed by specialists to ensure the usefulness of information and insights.” – by the request of Academic Editor the medical parts of the introduction section has been reduced.
- The abstract was improved.
- The contributions of each author are described in section “Author Contributions” as it is demanded by MDPI template.
- After reading the comments from reviewers and Academic Editor, we understood that the goal of our work was not stated clearly enough. We corrected it in the introduction. The formulation of our goal is: “This work aims to quantify the amount of information stored in human-interpretable features computed from the image with lymphocytes obfuscated. This work aims not to achieve the best classification results but to determine how well the classification can be performed using the limited amount of information while remaining human-interpretable. In our previous article, the Deep Neural Networks were applied to the raw images to perform the classification. Although the results were satisfactory, the inherent black-box nature of Deep Neural Networks prevented us from acquiring human-interpretable knowledge on the nature of this particular classification problem. This work aims to provide such information.”. Our goal was not to obtain the best possible classification quality but to determine the impact of a supposedly classification-neutral background and to obtain high-quality classification while using only human-interpretable features of the image. Covering lymphocytes was not to obtain better results but to determine how good the indicator of Leukemia is surrounding the lymphocyte. This work is the continuation of our previous work on this dataset titled: “IoT application of transfer learning in hybrid artificial intelligence systems for acute lymphoblastic leukemia classification”, so our main point of reference was our previous work, and the rest of them is to describe state-of-the-art methods in Leukemia classification better. We added the results of comparison between our previous work and current in the discussion.
- As stated in the article, this work is a continuation of our article: “IoT application of transfer learning in hybrid artificial intelligence systems for acute lymphoblastic leukemia classification”. That particular work was conducted solely on the ALL-IDB 2 dataset. This article is a deeper analysis of patterns present in the dataset with an additional focus on the human interpretability of extracted features. We decided to perform research on the ALL-IDB 2 dataset to compare the results with those previously obtained and conclude whether the application of Deep Neural Networks containing millions of parameters is viable compared to much simpler models. The discussion was extended to signify the continuation of the research better.
- The Discussion and Conslusions sections were expanded.
- We added the comparison between our previous work and the current one. The best method improved accuracy from 94.8% to 96.0%. We presented only the metrics of accuracy due to space restrictions. The work already contains four big tables with just accuracies. However, if more metrics are required, then we can add them. We also disagree with the statement: “The experimental results obtained from the proposed model evaluation seem not attractive.”. Our simple, computationally-effective, human-interpretable solutions obtained results up to 96.0% accuracy averaged from 30 experiments. It is not only a pretty good result but also obtained by, as mentioned before, human-interpretable feature extraction compared to other state-of-the-art black-box convolutional neural network features.
- Unfortunately, the reference order of appearance is managed by the LaTeX template. We improved the quality of the text.
Round 2
Reviewer 1 Report
I am particularly pleased with the responses and the improvement of the manuscript which is now much clearer.
regarding my question number 6, starting with line 356 "The metrics used to measure the quality of classification are accuracy and F1 score. ...' are not a result, but the definition of the metrics, i.e. the previous section.
Author Response
Thank you very much for your insight. We moved "Metrics" to the materials and methods section, which is currently on line 343.
The latexdiff tool for creating difference files creates UI issues like overflowing text to line numeration. This issue is related solely to the difference version and will not be present in the final version.
Reviewer 2 Report
The authors removed or corrected all specified shortcomings. In my opinion, the manuscript is good enough for publication.
Author Response
Thank you very much for your acceptance and your insight into the process of this work betterment.
Reviewer 3 Report
The authors have responded to some of my comments. The following comments are not yet satisfied:
The summary of contributions is not provided explicitly. The authors need to provide a subsection to summarize the list of contributions of their current research. Please check the following example:
https://www.mdpi.com/2079-9292/11/4/556/htm
The Literature Review section is not well developed. I suggest adding a comparative table toward the end of ''Introduction '' to contrast your solution with other state-of-art-related works. Please check the following example:
https://www.mdpi.com/2079-9292/11/4/556/htm
The future directions of the authors’ contribution are not provided. I suggest that the authors state some explicit future recommendations to improve or upgrade their model.
The experimental results obtained from the proposed model evaluation seem not attractive. However, the authors must provide a comparison with other state-of-the-art models in the same area of study. The comparison should consider all relevant performance factors such as accuracy, prediction time, precision,..etc.
Weak English. The authors need to proofread the paper to avoid any typos. Make sure that all references are cited in order of their appearance.
Author Response
Thank you for your insight into how we can improve our work. We did our best in the process of this article's betterment.
Before we address your comments, we would like to point out some graphical issues in the diff version uploaded. We used latexdiff tool for file comparison. It turns out that this tool generated some UI issues, like text overflowing into the line numeration. This is an issue regarding only the diff version; it will not be present in the final version.
- The summary of contribution was added on line 131
- A comparative table was added on line 108
- The future directions were added on line 485
- The comparison was added on line 392 and extended the paragraph from line 443
- We fixed all the typos we found. Unfortunately, we cannot control the order of citations as the MDPI Latex template governs it
Round 3
Reviewer 3 Report
Thank you, authors, for responding to my comments. However, as I mentioned earlier, the experimental results obtained from the proposed model evaluation seem not attractive. In your state of art comparison, you only compared your work with reference 35 which has better performance factors than what you did.
Also, the authors did not provide a convincing justification for the results obtained.
Therefore, I can not accept this paper as other state-of-the-art models already have better performance factors that are much satisfied.